# Glyceraldehyde-3-Phosphate Dehydrogenase Increases the Adhesion of *Lactobacillus reuteri* to Host Mucin to Enhance Probiotic Effects

**DOI:** 10.3390/ijms21249756

**Published:** 2020-12-21

**Authors:** Zhaoxi Deng, Tian Dai, Wenming Zhang, Junli Zhu, Xin M. Luo, Dongyan Fu, Jianxin Liu, Haifeng Wang

**Affiliations:** 1College of Animal Science, Zhejiang University, Hangzhou 310058, China; zhaoxideng@zju.edu.cn (Z.D.); 15957132027@163.com (T.D.); rebelwenming@163.com (W.Z.); 21817060@zju.edu.cn (D.F.); liujx@zju.edu.cn (J.L.); 2College of Food Science and Biotechnology, Zhejiang Gongshang University, Hangzhou 310012, China; jlzhu0305@mail.zjgsu.edu.cn; 3Department of Biomedical Sciences and Pathobiology, Virginia Tech, Blacksburg, VA 24060, USA; xinluo@vt.edu

**Keywords:** *Lactobacillus reuteri*, GAPDH, adhesion, mucin, intestine

## Abstract

The ability to adhere to the intestinal mucus layer is an important property of probiotic bacteria. *Lactobacillus reuteri* strains ZJ615 and ZJ617 show low and high adhesion, respectively, to intestinal epithelial cells. In this study, we quantified bacterial cell wall-associated glyceraldehyde-3-phosphate dehydrogenases (cw-GAPDH) and bacterial cell membrane permeability in both strains using immunoblotting and flow cytometry, respectively. Highly adhesive *L. reuteri* ZJ617 possessed significantly more cw-GAPDH, higher cell membrane permeability, and significantly higher adhesive ability toward mucin compared with low-adhesive *L. reuteri* ZJ615. In vitro adhesion studies and analysis of interaction kinetics using the Octet, the system revealed significantly decreased interaction between *L. reuteri* and mucin when mucin was oxidized when bacterial surface proteins were removed when bacteria were heat-inactivated at 80 °C for 30 min, and when the interaction was blocked with an anti-GAPDH antibody. SWISS-MODEL analysis suggested intensive interactions between mucin glycans (GalNAcα1-O-Ser, GalNAcαSer, and Galβ3GalNAc) and GAPDH. Furthermore, in vivo studies revealed significantly higher numbers of bacteria adhering to the jejunum, ileum, and colon of piglets orally inoculated with *L. reuteri* ZJ617 compared with those inoculated with *L. reuteri* ZJ615; this led to a significantly decreased rate of diarrhea in piglets inoculated with *L. reuteri* ZJ617. In conclusion, there are strong correlations among the abundance of cw-GAPDH in *L. reuteri*, the ability of the bacterium to adhere to the host, and the health benefits of this probiotic.

## 1. Introduction

Probiotics are a class of microorganisms that exert beneficial effects on host health by reducing or preventing gastrointestinal disorders such as diarrhea, irritable bowel syndrome, and inflammatory bowel disease [1]. In the host intestine, a dense mucus layer prevents inflammation by shielding the underlying epithelium from luminal bacteria and food antigens [2]. The ability of probiotic bacteria to adhere to the intestinal mucus layer, a prerequisite for bacterial colonization, is often considered an important criterion in their selection for commercial use [3]. Adherence of lactobacilli to the intestinal mucosal layer is mediated by surface proteins with a mucus-binding capacity [4,5]. Well-known mucus-binding proteins adhering to the outer mucus layer include the canonical mucus-binding protein (MUB) in *Lactobacillus reuteri* ATCC 53608 cell, mucus binding protein A (CmbA) and sortase-dependent proteins (SDPs) in *L. reuteri* ATCC PTA 6475, the multi-repeat cell-surface adhesin Lar0958 in *L. reuteri* JCM 1112T, surface proteins (42 and 114 kDa) in *Lactobacillus rhamnosus* PEN, and the mucus-binding pili in strains of *L. rhamnosus* [6,7,8,9,10,11].

Adhesion of lactobacilli to mucin is one of the critical factors contributing to the persistent beneficial effects of probiotic bacteria in a constantly changing intestinal environment [6]. Indeed, adhesion of lactobacilli to the mucus often provides these bacteria with a competitive advantage over pathogens within the gut ecosystem [12]. Mucins harbor glycan-rich domains that allow the binding of pathogens and commensal bacteria [13]. Mucins are rich in Ser/Thr residues that are targets for *O*-glycosylation, primarily with *N*-acetylgalactosamine (GalNAc), creating the foundation upon which long and more complex oligosaccharide chains are built. The *O*-glycan chains are extended by sequential addition of monosaccharides such as N-acetylglucosamine (GlcNAc), galactose (Gal), GalNAc, fucose (Fuc), and sialic acid (*N*-acetylneuraminic acid; Neu5Ac) [14]. The glycan structures present in mucins are diverse and complex and consist of four core mucin-type *O*-glycans containing GalNAc, Gal, and GlcNAc [15]. Diverse glycan structures along the gastrointestinal tract provide binding sites for the gut bacteria that have adapted to the mucosal environment by expressing the appropriate adhesion components [16]. Some sugar-binding proteins in gut bacteria and interactions of the sugar-binding proteins with mucus have been structurally characterized [17]. However, the mechanism by which lactobacilli surface proteins interact with mucins is unclear [6,18]. Purified mucins immobilized on microtiter plates or diagnostic glass slides have been extensively used to quantitatively evaluate *Lactobacillus* adhesion [19].

Glyceraldehyde-3-phosphate dehydrogenase (GAPDH) is a cytoplasmic protein involved in glycolysis but has many other physiological functions [20,21]. Similar to other secretory glycosylases, GAPDH can re-ligate to the cell wall as a non-anchored protein after secretion and acts as a moonlighting protein during adhesion of lactobacilli to mucin [22,23,24,25]. *L. reuteri* ZJ617 and *L. reuteri* ZJ615 were isolated from porcine intestines in our previous study and were designated as high- and low-adhesive strains, respectively [26]. Indirect immunofluorescence and immunoelectron microscopy methods localize GAPDH to the surface of *L. reuteri* ZJ617 [26]. This GAPDH is associated with the bacterial cell wall (and is thus termed cw-GAPDH) and acts as an adhesion component in the binding of *L. reuteri* ZJ617 to intestinal epithelial cells [27]. However, it is not known whether the cell surface abundance of cw-GAPDH is associated with the adhesive ability of lactobacilli, how GAPDH is secreted from the cytoplasm to the outer cell wall (namely, transmembrane secretion), or how GAPDH is distributed on the bacterial cell wall.

The objectives of this study were to investigate the secretion and distribution of GAPDH on the cell surface of *L. reuteri* strains with different adhesion abilities and delineate the function of GAPDH in mediating bacterial adhesion to mucin. We employed an improved twin-probe multiparameter flow cytometric technique to examine cell membrane permeability with fluorescein diacetate (FDA, green) and propidium iodide (PI, red), two fluorescent dyes that indicate intact or permeable membranes, respectively [28,29,30]. We further characterized the mechanism of interaction between mucin glycans and GAPDH using SWISS-MODEL, a pioneering tool in the field of automated protein modeling [31].

## 2. Results

### 2.1. Subsection Dynamics of cw-GAPDH on the Surface of L. reuteri Strains with Different Adhesion Abilities

High-adhesive *L. reuteri* ZJ617 had more cw-GAPDH on the bacterial cell surface than low-adhesive *L. reuteri* ZJ615 when cultures of the two strains were adjusted to the same concentration (Figure 1A). The amount of cw-GAPDH on the bacterial cell surface increased significantly with bacterial growth (logarithmic growth phase) from 3 to 6 h and was highest at 6 h for both strains, decreasing significantly at 9 h and during the subsequent late growth stage at 12, 15, and 18 h (*p* < 0.05, Figure 1B).

### 2.2. Time Course of Cell Membrane Permeability of L. reuteri Strains with Different Adhesion Abilities

Cells of *L. reuteri* ZJ615 and *L. reuteri* ZJ617 were collected at different growth stages (3, 6, 9, 12, 15, 18 h), labeled with FDA/PI, and analyzed by flow cytometry (Figure 2). The percentage of FDA-/PI+ and FDA+/PI− cells was 64.20% and 1.72%, respectively, for *L. reuteri* ZJ615 cultured for 3 h, whereas by 18 h, this had changed to 0.68% and 88.30%, respectively (Figure 2A). This indicated that cell membrane permeability had decreased between 3 h and 18 h. A similar trend was observed for *L. reuteri* ZJ617 (Figure 2B); cell membrane permeability was highest at 6 h (*p* < 0.05), with 84.30% FDA−/PI+ cells and 0.03% FDA+/PI− cells. The cell membrane permeability of *L. reuteri* ZJ617 then decreased with time (9 and 12 h) and reached its lowest point at the late-stage of bacterial growth (15 and 18 h) (*p* < 0.05, Figure 2C). High-adhesive *L. reuteri* ZJ617 had significantly higher cell membrane permeability than low-adhesive *L. reuteri* ZJ615 at multiple time points (3, 6, 9, and 12 h) (*p* < 0.05, Figure 2C). The correlation index between the abundance of cw-GAPDH and cell membrane permeability was 0.67 (Figure 2D) and 0.80 (Figure 2E) for *L. reuteri* ZJ615 and *L. reuteri* ZJ617, respectively, with a correlation index of 0.53 for both strains when data were combined (Figure 2F).

### 2.3. Distribution of GAPDH in Lactobacilli Strains

Immunoelectron microscopy images showed that the distribution of GAPDH on the surface and cytoplasm of *L. reuteri* ZJ615 and ZJ617 was different. (Figure 3). Longitudinal and trans-section images of *L. reuteri* ZJ615 and *L. reuteri* ZJ617 collected at 6, and 15 h showed GAPDH localized on the surface and in the cytoplasm of both *L. reuteri* ZJ615 (Figure 3A) and ZJ617 (Figure 3B). However, the amount of GAPDH localized on the surface and in the cytoplasm of *L. reuteri* ZJ617 was generally higher than that observed for *L. reuteri* ZJ615 collected at the same time points. Between the two time points, a greater amount of GAPDH was found on the surface of *L. reuteri* ZJ615 (Figure 3A) or *L. reuteri* ZJ617 (Figure 3B) at 6 h compared to 15 h.

### 2.4. Adhesion of the Two L. reuteri Strains to Mucin

Adhesion of both *L. reuteri* ZJ615 and *L. reuteri* ZJ617 to mucin decreased significantly after treatment with an anti-GAPDH antibody, after mucin oxidation by periodate, and after both these treatments (*p* < 0.05, Figure 4A,B). Notably, for untreated samples and samples treated with both an anti-GAPDH antibody and periodate, the levels of adhesion to mucin were significantly higher for *L. reuteri* ZJ617 than for *L. reuteri* ZJ615 (*p* < 0.05, Figure 4B).

### 2.5. Dynamic Interaction between the Two Strains of L. reuteri and Mucin

Time-resolved dynamic interaction between the individual strains of *L. reuteri* and mucin was evaluated using an Octet system to observe association and disassociation processes. For both strains of *L. reuteri*, the interaction with mucin decreased significantly in the presence of an anti-GAPDH antibody (Figure 5A). Significantly reduced interactions were also observed when the lactobacilli strains were treated with lithium chloride (LiCl) to remove surface proteins including GAPDH (Figure 5B), when the bacteria were inactivated by heating at 80 °C for 30 min (Figure 5C), and when the mucin was treated with periodate to oxidize glycans (Figure 5D). In addition, with the exception of heat inactivation, high-adhesive *L. reuteri* ZJ617 had a significantly more intense interaction with mucin than low-adhesive *L. reuteri* ZJ615, regardless of treatment.

### 2.6. Molecular Interaction between GAPDH and Mucin Glycans

According to our previous results, the protein sequence of GAPDH deduced from the genomic sequence of *L. reuteri* ZJ617 has 335 amino acids [27]. SWISS-MODEL modeled 312 residues (93% of the amino acid sequence) with 100.0% confidence using the single highest scoring template for GAPDH (Figure 6A). Mucin glycans enter the active pockets of GAPDH and interact through hydrogen bonds. Within the *L. reuteri* ZJ617 GAPDH model, there were 9, 7, and 11 active interaction sites for GalNAcα1-O-Ser (Figure 6B), GalNAcαSer (Figure 6C), and Galβ3GalNAc (Figure 6D), respectively (Table 1). A docking box was set up and used to cover all the binding site residues. All ligands, including GalNAcα1-O-Ser (Figure 6B), GalNAcαSer (Figure 6C), and Galβ3GalNAc (Figure 6D), rigidly docked into the binding pocket; detailed 3D interactions of GAPDH in complex with the glycans are presented (Figure 6B–D). The binding energy values of GalNAcα1-O-Ser, GalNAcαSer, and Galβ3GalNAc with GAPDH were −3.43, −3.24, and −3.33 kcal/mol, respectively (Table 1). The most common amino acids exposed on the surface of GAPDH that formed hydrogen bonds with these glycans were Met1, Ser2, Val3, Ser29, Asp30, and Ile31 (Table 1).

### 2.7. Adhesion of the Two L. reuteri Strains in the Porcine Intestine

Compared with *L. reuteri* ZJ615, significantly more count of *L. reuteri* ZJ617 was excreted in feces on days 13, 17, and 25 (Figure 7A). In addition, bacterial numbers recovered from the jejunum, ileum, and colon (i.e., adherent bacteria) were significantly greater in pigs orally inoculated with *L. reuteri* ZJ617 than in pigs inoculated with *L. reuteri* ZJ615 (Figure 7B). Importantly, piglets orally inoculated with high-adhesive *L. reuteri* ZJ617 had a significantly lower diarrheal rate than piglets inoculated with low-adhesive *L. reuteri* ZJ615 (*p* < 0.05, Figure 7C).

## 3. Discussion

The number of *L. reuteri* ZJ617 is significantly higher than that of *L. reuteri* ZJ615 in both intestines and feces. This may indicate that *L. reuteri* ZJ617 has a higher ability to colonize the intestines and more efficiently reproduce there. This is also consistent with the fact that *L. reuteri* ZJ617 and ZJ615 were isolated from porcine intestines as strains with the high- and low-adhesive ability to intestinal epithelial cells, respectively [26]. In mouse intestines, *L. reuteri* ZJ617 also has a higher adhesive ability than *L. reuteri* ZJ615 [32]. High-adhesive *L. reuteri* ZJ617 showed a significantly higher abundance of cw-GAPDH than low-adhesive *L. reuteri* ZJ615, which may explain the difference in adhesion between the two strains.

GAPDH possesses plasma membrane-associated functions and exhibits enzymatic activities different from those of cytoplasmic GAPDH when it reaches the cell surface as a moonlighting protein [33,34]. Removal of surface proteins such as GAPDH, elongation factor Tu (EF-Tu), and triosephosphate isomerase (TPI) by guanidine hydrochloride decreases the ability of lactobacilli to adhere to Caco-2 human colon carcinoma cells by 40%, suggesting that these surface proteins are important for bacterial adhesion to human intestinal cells [35]. GAPDH is present as an anchorless molecule on the surface of *L. plantarum* 299v and *L. crispatus* and is involved in bacterial adhesion through anon-glycolytic moonlighting function [36,37]. In addition, *L. plantarum* LA 318 isolated from normal human intestinal tissues has a high level of adhesion to human colonic mucin, a process mediated by GAPDH on the bacterial cell surface [38]. Moreover, an anti-GAPDH antibody significantly reduces the adhesion of *L. reuteri* to intestinal epithelial cells [27]. Together, these data suggest that bacterial GAPDH plays an important role in the adhesion of the bacteria to the host. Consistent with this notion, we demonstrated that significantly more *L. reuteri* ZJ617 than *L. reuteri* ZJ615 adheres to the porcine intestine, and this may result, at least in part, from *L. reuteri* ZJ617 possessing significantly more cw-GAPDH than *L. reuteri* ZJ615.

Indirect immunofluorescence analysis has shown that GAPDH is distributed on the surface of *L. crispatus* [37]. For GAPDH to be exposed on the surface of lactobacilli, the bacterial cell membrane needs to be permeable. Here, secretion of GAPDH peaked in the logarithmic growth phase (6 h). Bacterial cell membrane permeability was also highest at this time, indicating that an increase in cell membrane permeability contributes to the secretion of GAPDH. However, another study showed that the cw-GAPDH activity of *L. plantarum* 299v increased significantly from the incubation period to the stationary period [39]. This is inconsistent with our results, probably because we adjusted the number of cells at different time points to 10^8^ cells/mL. However, we came to a consistent conclusion that an increase in the concentration of cw-GAPDH was closely related to an increase in plasma membrane permeability during growth. Furthermore, immune electron microscopy revealed differences in the abundance of GAPDH between the high-and low-adhesive strains of *L. reuteri*. Notably, high-adhesive *L. reuteri* ZJ617 had significantly higher cell membrane permeability and more surface GAPDH compared with low-adhesive *L. reuteri* ZJ615.

Mucin functions as an adhesion receptor for surface GAPDH in lactobacilli [38,40]. Adhesion of lactobacilli to the intestine is partly due to the binding of GAPDH to human ABO-type blood group antigens expressed on human colonic mucin [41]. In addition, GAPDH localized on the surface of *Mycoplasma genitalium* is involved in binding to mucin [42]. In this study, the number of lactobacilli adhering to porcine intestinal mucin was significantly decreased in the presence of an anti-GAPDH antibody, indicating that cw-GAPDH plays an important role in the adhesion of lactobacilli to mucin. The number of lactobacilli adhering to mucin was also significantly decreased after mucin was oxidized by periodate. Although adhesion abilities to normal mucin differ between *L. reuteri* ZJ617 and ZJ615, both strains adhered weakly to oxidized mucin, suggesting that periodate oxidizes mucin glycosylation and destroys the site interacting with GAPDH. Consistent with this, the ability of *L. plantarum* LA 318 to adhere to human colonic mucin is significantly decreased following periodate-mediated oxidation of mucin [41].

We used the Octet system to determine in vitro time-resolved interaction dynamics between lactobacilli and mucin, revealing a high affinity between the two. High-adhesive *L. reuteri* ZJ617 showed a significantly higher affinity for mucin compared with low-adhesive *L. reuteri* ZJ615. Interactions between mucin and both strains of *L. reuteri* decreased significantly after mucin glycans were oxidized and removed, or GAPDH was removed or blocked, providing additional evidence for the high affinity between GAPDH and mucin. Moreover, size exclusion chromatography has shown that the 39-kDa GAPDH from *Lactobacillus acidophilus* (r-LaGAPDH) exists as a tetramer in solution and has mucin-binding and hemagglutination activities, suggesting a carbohydrate-mediated adhesion mechanism [43].

Notably, some lactobacilli still adhered to mucin even in the presence of an anti-GAPDH antibody. This suggests that other related molecules may have similar functions during colonization by lactobacilli. Such molecules could include adhesions such as mucus-binding protein (Mub), elongation factor Tu (EF-Tu), lipoteichoic acid (LTA), and extracellular polysaccharides [10,44]. Mub is unique to gut-resident lactic acid bacteria and plays an important role in host-microbial crosstalk and pathogen exclusion [7]. In *L. plantarum*, this activity is mediated through the final two domains (Mubs5s6) of the six mucus-binding domains arranged in tandem at the C-terminus of the Lp_1643 (Mub) protein.

Human GAPDH is a 36-kDa polypeptide with 335 amino acids. It is homotetrameric and best described as a dimer of dimers [33]. The 3D structures of bacterial and eukaryotic GAPDH proteins are very similar, sharing high sequence similarity and both using NAD^+^ as a cofactor [33]. The GAPDH homotetramer possesses two 10-Å long grooves along the P interface that encompass the binding regions [45]. GAPDH of *L. reuteri* ZJ617 also has 335 amino acids and comprises α-helix, β-sheet folding, and sheet structures. These spatial structures form the groove containing adhesion sites. Binding of full-length *L. reuteri* Mub to mucus involves multiple interactions mediated by terminal sialylated mucin glycans [9].

Molecular docking analysis of the interaction between *L. reuteri* GAPDH and mucin glycans supported the in vitro time-resolved interaction dynamics data. We found similar binding energy values (kcal/mol) between GAPDH and the glycans GalNAcα1-O-Ser (−3.43), GalNAcαSer (−3.24), and Galβ3GalNAc (−3.33). The lower the binding energy, the more stable the ligand–protein complex. Notably, *L. reuteri* GAPDH could bind different mucin glycan molecules, suggesting the presence of multiple glycan-binding sites. GAPDH from lactobacilli can bind human blood group A- and B-antigens [41]. In contrast, EF-Tu interacts with a sulfated galactose residue of mucin, whereas the mucin-binding domain (MucBD) recognizes terminal sialic acid and sulfated carbohydrate moieties [18]. In addition, a 93-amino acid mucin-binding domain (MBD93) from *Lactobacillus fermentum* interacts strongly with terminally expressed mucin glycans including GalNAc, GlcNAc, Gal, and sialic acid through its surface amino acid residues, Ser^57^, Pro^58^, Ile^60^, Tyr^63^, and Ala^65^, a process that involves hydrogen bonds [46]. Homology modeling followed by docking analysis revealed that GAPDH of *L. reuteri* interacts with mucin through the key surface residues Met^1^, Ser^2^, Val^3^, Ser^29^, Asp^30^, and Ile^31^. Importantly, our docking studies supported the results of in vitro Octet experiments indicating that *L. reuteri* GAPDH interacts with mucin glycans. Future studies will involve more detailed structural analyses to elucidate how the cw-GAPDH of lactobacilli interacts with mucin.

In conclusion, high-adhesive *L. reuteri* ZJ617 possesses significantly more cw-GAPDH than low-adhesive *L. reuteri* ZJ615, and this correlates with increased bacterial cell membrane permeability. Interaction between the cw-GAPDH of lactobacilli and host mucin may play an important role in bacterial adhesion to the intestine, thus mediating the health benefits of lactobacilli.

## 4. Materials and Methods

### 4.1. Bacterial Strains and Growth Conditions

*L. reuteri* ZJ617 and ZJ615 were isolated from the intestine of piglet and kept in our laboratory. Both strains were cultured in de Man-Rogosa-Sharpe (MRS) broth anaerobically at 37 °C and collected at 3, 6, 9, 12, 15, and 18 h after inoculation.

### 4.2. Quantification of cw-GAPDH on Lactobacilli by Immunoblotting

By drawing growth curves at different time points, and cultures were adjusted to an OD600 of 1.9 by dilution or concentration. Moreover, surface proteins were extracted with 5 M LiCl from lactobacilli harvested at different time points as previously described [47]. Immunoblotting analysis was performed as previously described [27]. The primary antibody was rabbit anti-GAPDH generated in our previous study (Sangon Biotech, Shanghai, China; 1:1000) [27]. The specific proteins were detected by using an enhanced chemiluminescence kit (Perkin Elmer Life Sciences, Boston, MA, USA). Protein bands were visualized with a chemiluminescent substrate and a gel-imaging system (Tanon Science and Technology, Shanghai, China) and analyzed via the Image Analysis Software (National Institutes of Health, Bethesda, MD, USA).

### 4.3. Analysis of Membrane Permeability

*L. reuteri* ZJ617 and ZJ615 were cultured in MRS broth at 37 °C for 3, 6, 9, 12, 15, and 18 h. Bacteria were harvested by centrifugation (4000× *g* for 5 min at 10 °C) and washed twice with PBS. Bacterial suspensions in PBS were concentrate or diluted to a concentration of 10^8^ cells/mL. FDA (Sigma, Darmstadt, Germany) was dissolved in acetone and adjusted to a concentration of 5 mg/mL. PI (Sigma, Darmstadt, Germany) was dissolved in Milli-Q water and adjusted to a concentration of 1 mg/mL. Lactobacilli in suspension were incubated with FDA at 4 °C for 20 min, then with PI at 4 °C for 5 min in a darkroom [39]. Labeled bacteria were kept on ice prior to FCM analysis. All FCM were carried out using a Facsverse flow cytometer (Becton Dickinson, Franklin Lakes, NJ, USA) at 488 nm excitation using a 25 mW Argon laser with a 70 μm sense flow cell. Cell samples were delivered at a low flow rate of 500 cells/s. At least 100,000 cells were analyzed for each sample. A 530 nm bandpass filter and a 660 nm long pass filter were used to collect the green fluorescence (FL1) and the red fluorescence (FL3), respectively. The labeling with each fluorochrome was done so that each measure of fluorescence was independent of the others. Bacteria killed by heat inactivation at 80 °C for 30 min and stained only with PI were used as a negative control. Data were analyzed with the Cell Quest software (Becton Dickinson, Franklin Lakes, NJ, USA).

### 4.4. Immunoelectron Microscopy

GAPDH on the surface and in the cytoplasm of *L. reuteri* ZJ617 and ZJ615 cultured for 6 and 15 h was examined by immunoelectron microscopy according to a previously published method [17,27]. *Lactobacillus reuteri* was washed with PBS three times, then mixed with 0.5% (m/v) agarose agar and changed into a solid state. The sample was fixed and dehydrated with a series of gradient dilution ethanol, then embedded into K4M resin. After polymerization, ultrathin sections were cut and collected onto carbon-coated 200-mesh nickel grids. The sections were incubated with a rabbit anti-GAPDH antibody and Protein A-gold (10 nM in size, Sigma, Darmstadt, Germany). After being washed with PBS, the sections were post stained in uranyl acetate and lead citrate before examination with transmission electron microscopy (TEM) (JEM-1010, JEOL Company, Tokyo, Japan).

### 4.5. Microscopic Observation of the Adhesion of Lactobacilli to Mucin

*L. reuteri* ZJ617 and ZJ615 were cultured for 18 h, harvested by centrifugation (4000× *g* for 5 min at 10 °C) and resuspended in PBS at 1 × 10^8^ colony-forming units (CFU)/mL. Glass slides were coated with porcine intestinal mucin (Sigma, Darmstadt, Germany) overnight, dried and fixed with formaldehyde for 20 min. Placed in a Petri dish containing 20 mL bacterial suspension (one slide per dish), and incubated for 1 h at 37 °C. Slides were then removed and washed five times with sterile physiological saline to remove non-adherent bacteria. Adherent bacteria were counted under a microscope after Gram staining [26]. *L. reuteri* cells were blocked with an anti-GAPDH antibody, as previously described, or mucin was oxidized by treatment with periodate [27].

### 4.6. Kinetic Analysis of Lactobacilli Adhesion to Mucin

*L. reuteri* ZJ617 and ZJ615 were grown anaerobically in MRS at 37 °C for 18 h. Bacteria were then killed by heat treatment at 85 °C for 30 min, treated with LiCl to remove the surface layer protein, or blocked using an anti-GAPDH antibody according to a previously published method [26]. Bacteria with or without treatment were suspended in PBS to a concentration of 10^8^ CFU/mL and immobilized on an amino propylsilane (APS, Phoenix, AZ, USA) sensor (ForteBio, Fremont, CA, USA) for 3 min. Porcine intestinal mucin, with or without periodate oxidation, was used as the mobile phase at a concentration of 330 nM. The analysis was carried out using an Octet apparatus and its software (Pall ForteBio, Menlo Park, CA, USA). Association and dissociation processes between bacteria and mucin were measured in PBS for 5 min [48].

### 4.7. Prediction of Mucin-Binding Sites on GAPDH

The amino acid sequence of GAPDH from *L. reuteri* was deduced from the genomic sequence [27]. A template search of GAPDH was performed and evaluated using SWISS-MODEL (https://swissmodel.expasy.org/). Templates with the highest scoring crystal structure were selected for model building according to the target-template alignment using ProMod3 [49]. Three-dimensional models of ligands GalNAcα1-O-Ser, GalNAcαSer, and Galβ3GalNAc were generated using the Pubchem database (https://pubchem.ncbi.nlm.nih.gov/) to form the PDB model data [50].

AutoDock (https://adfr.scripps.edu/) uses a Lamarckian genetic algorithm to calculate the possible conformations in which a ligand binds to a protein. Flexible ligands were set to be free in the binding clefts of GAPDH to dock in the most feasible way. The best-docked structures of GAPDH ligands based on binding energy scores were chosen for further analyses. The unbound interaction and binding energy (in kcal/mol) were analyzed to assess the strength of binding. A file was procured with .ent extension, which was manually saved in the .pdbqt format for docking. The molecular docking simulation was performed using AutoDock 4.2.6 embedded in AutoDockTools-1.5.6. Water molecules were deleted, and polar hydrogens were added to all ligands and proteins using the AutoDockTools program prior to docking using the Autodock 4.2.6 program, where the prediction of ligand-binding sites in the modeled protein structure was made [51]. Hydrogen bonds and hydrophobic interactions between the ligand and the protein were calculated using ligplot+1.4 (https://www.ebi.ac.uk/thornton-srv/software/LigPlus/), whereas a space-filling model was generated using PyMol 2.2.0 (PyMOL Molecular Graphics System, New York, USA).

### 4.8. Adhesion of Lactobacilli in the Porcine Intestine

*L. reuteri* ZJ617 and ZJ615 were screened for streptomycin resistance. Bacterial strains were resuspended in reconstituted skim milk (10%, w/v) and freeze-dried for 14 h in a vacuum freeze dryer. The resulting bacterial powder was stored in sealed packets at 4 °C until use. Twelve healthy 3-day-old Duroc × Landrace × Large White piglets with similar body weight (around 1.3 kg) were selected and divided into two equal groups. Oral inoculation of the piglets with *L. reuteri* ZJ615 and ZJ617 was conducted at 3–16 days of age, with a daily incremental dose increase (from 103 to 109 CFU/dose). All animal procedures were approved by the Animal Care and Use Committee of Zhejiang University (ETHICS CODE Permit No. ZJU20170529). Feces were collected on days 13, 17, and 25. All piglets were sacrificed on day 25, and jejunum, ileum, and colon were collected. The number of lactobacilli in discharged feces or homogenized intestinal segments (CFU/g sample) was determined by serial dilution on agar plates in the presence of streptomycin.

### 4.9. Statistical Analysis

Data were analyzed using SAS 9.1.3 (SAS Institute, Cary, NC, USA). Data are presented as means ± standard error of the mean (SEM). Means were compared by one-way ANOVA, followed by Duncan’s multiple range test. *p* value < 0.05 was considered significant. 

## Figures and Tables

**Figure 1 ijms-21-09756-f001:**
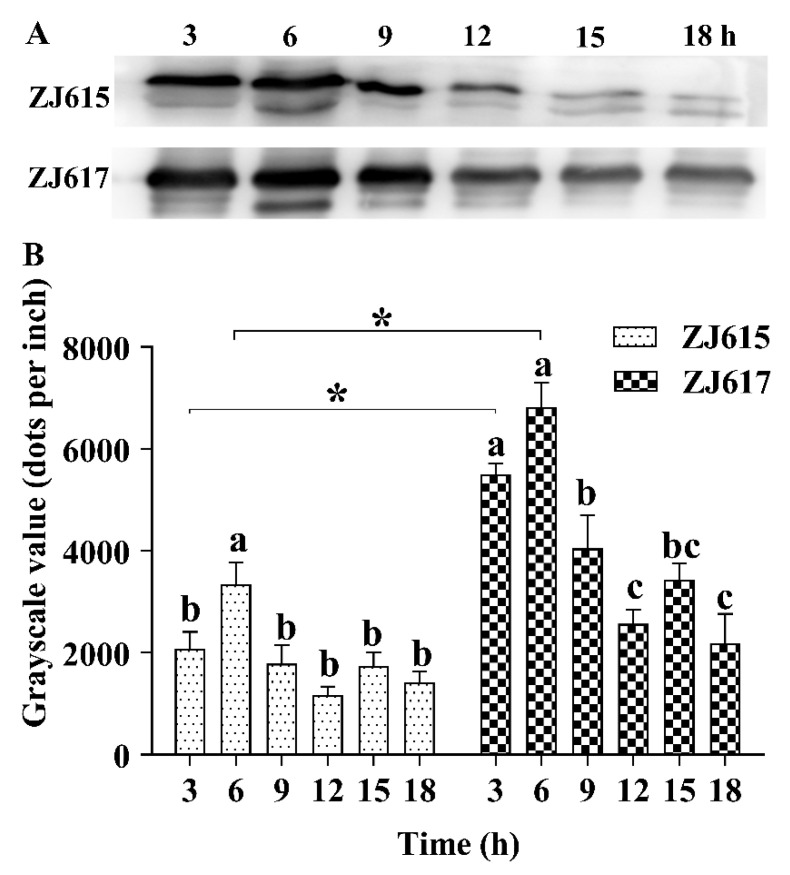
Immunoblot analysis of glyceraldehyde-3-phosphate dehydrogenase (GAPDH) on the surface of *L. reuteri* ZJ617 and ZJ615. Surface proteins were extracted, separated by dodecyl sulfate, sodium salt–polyacrylamide gel electrophoresis (SDS–PAGE) and transferred to a polyvinylidene fluoride (PVDF) membrane, which was then probed with a rabbit anti-GAPDH antibody. (**A**) Surface GAPDH proteins from *L. reuteri* ZJ617 and ZJ615. (**B**) Quantification of protein bands from the immunoblotting at different time points. Data are presented as mean + SD (*n* = 5). A one-way ANOVA with Duncan’s multiple range test was used. Different letters indicate significant differences within each strain (*p* < 0.05). Asterisks indicate significant differences between the two strains (*p* < 0.05).

**Figure 2 ijms-21-09756-f002:**
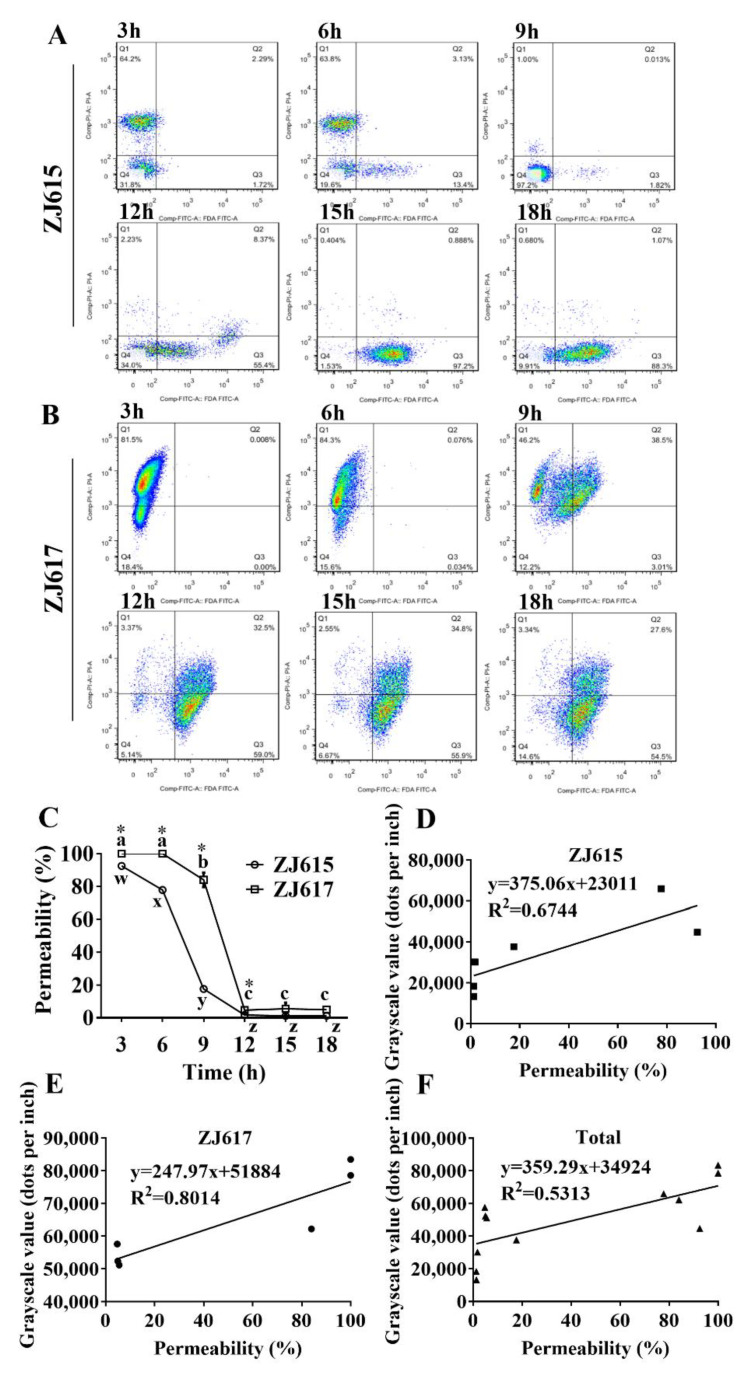
Cell membrane permeability of *L. reuteri* ZJ617 and ZJ615. (**A**,**B**) Bacteria were cultured in MRS at 37 °C for up to 18 h and separately stained with fluorescein diacetate (FDA) or propidium iodide (PI). Flow cytometry plots are shown for *L. reuteri* ZJ615 (**A**) and ZJ617 (**B**). (**C**) Cell membrane permeability of *L. reuteri* ZJ615 and ZJ617 cultured for 3, 6, 9, 12, 15, and 18 h. (**D**–**F**) Correlations between the amount of cw-GAPDH and the cell membrane permeability of *L. reuteri* ZJ615 (**D**), ZJ617 (**E**), and both strains together (**F**) (*n* = 5). A one-way ANOVA with Duncan’s multiple range test was used. Different letters indicate significant differences within each strain (*p* < 0.05). Asterisks indicate significant differences between the two strains (*p* < 0.05).

**Figure 3 ijms-21-09756-f003:**
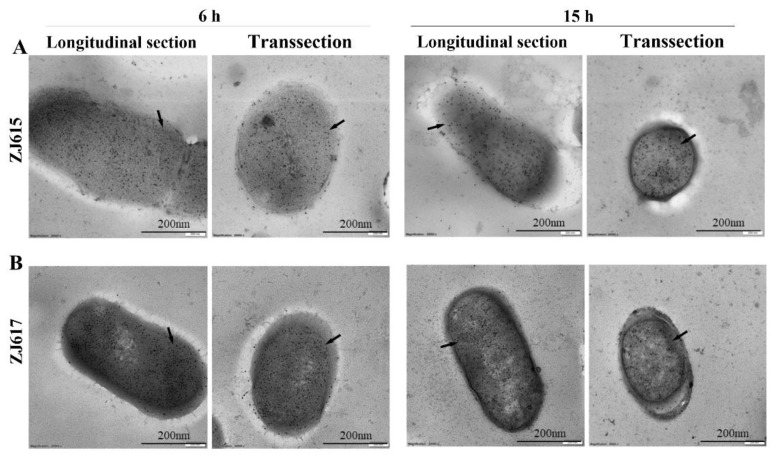
Immunoelectron microscopy analysis of GAPDH on the surface and in the cytoplasm of *L. reuteri* ZJ617 and ZJ615. GAPDH was visualized using an anti-GAPDH antibody and protein A-gold (10 nm-diameter particles) after bacteria were embedded and ultrathin sections were obtained. The black dots in the figure are labeled GAPDH protein, which is the position pointed by the arrow. (**A**) GAPDH in *L. reuteri* ZJ615 cultured for 6 h or 15 h. (**B**) GAPDH in *L. reuteri* ZJ617 cultured for 6 h or 15 h. Longitudinal (left) and trans-section (right) images are shown. Bars, 200 nm.

**Figure 4 ijms-21-09756-f004:**
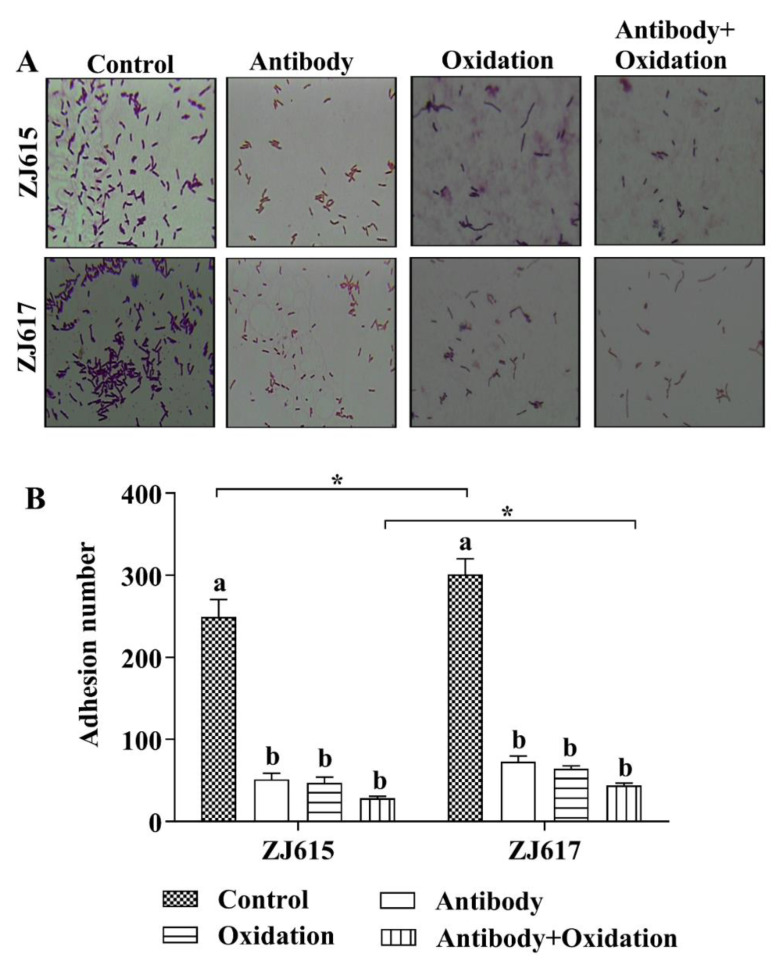
Adhesion of *L. reuteri* ZJ617 and ZJ615 to mucin. (**A**) Microscopy images of lactobacilli adhering to mucin-covered glass slides. Antibody bacteria were incubated with an anti-GAPDH antibody to block GAPDH activity. Oxidation, mucin was oxidized with periodate. (**B**) Quantification of the microscopy images (*n* = 5). Eight fields of vision were randomly selected under an oil microscope to calculate the number of bacteria adhered to the surface. Magnification = 1000×. A one-way ANOVA with Duncan’s multiple range test was used. Different letters indicate significant differences within each strain (*p* < 0.05). Asterisks indicate significant differences between the two strains (*p* < 0.05).

**Figure 5 ijms-21-09756-f005:**
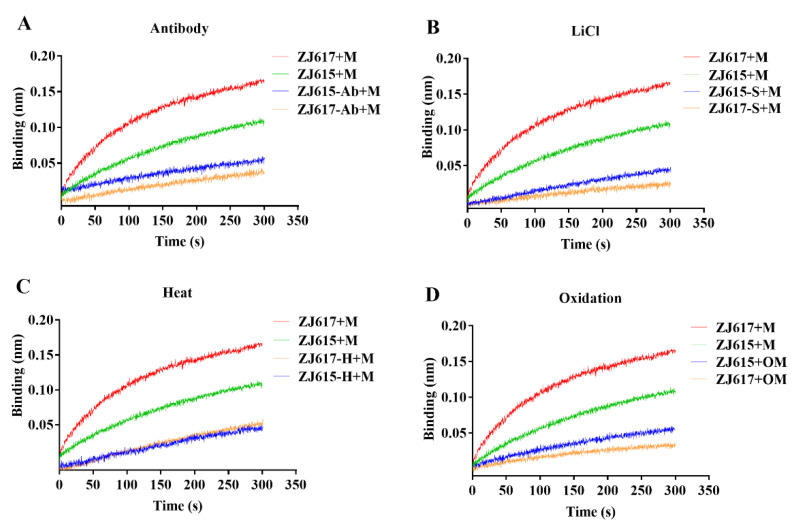
Interaction kinetics between *L. reuteri* ZJ617 and ZJ615 and porcine intestinal mucin determined using Octet apparatus. Bacteria were complexed on a chip, and mucin (M) was used as the mobile phase. (**A**) Kinetics when *L. reuteri* strains were blocked with an anti-GAPDH antibody (labeled “Ab”). (**B**) Kinetics when *L. reuteri* strains were treated with LiCl to remove surface proteins (labeled “S”). (**C**) Kinetics when *L. reuteri* strains were heat-inactivated at 80 °C for 30 min (labeled “H”). (**D**) Kinetics when mucin was oxidized with periodate (labeled “O”).

**Figure 6 ijms-21-09756-f006:**
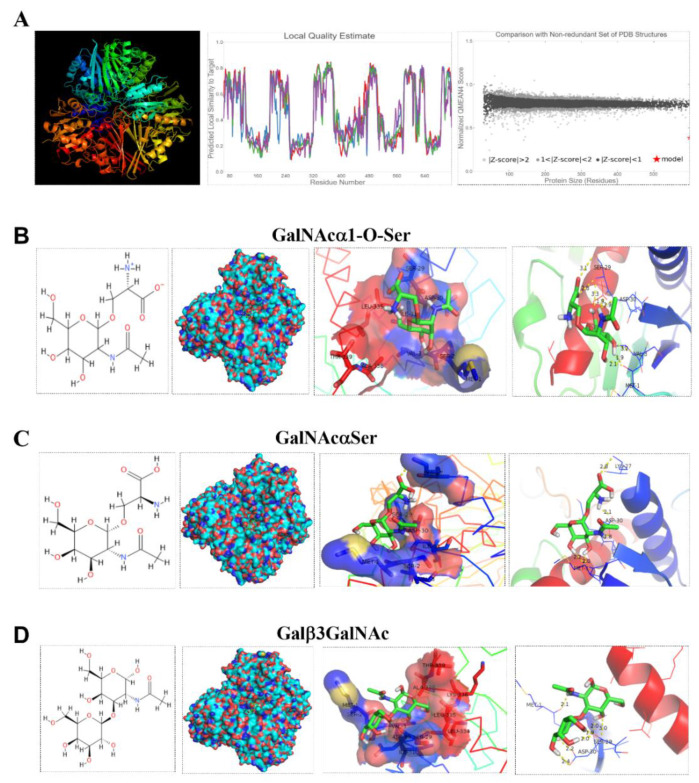
Molecular docking of mucinglycans with GAPDH. (**A**) Three-dimensional model generated by SWISS-MODEL. (**B**–**D**) Interactions of GalNAcα1-O-Ser (**B**), GalNAcαSer (**C**), and Galβ3GalNAc (**D**) docked to the homology model of *Lactobacillus* GAPDH. Interacting amino acid residues are labeled, and hydrogen bonds are shown as yellow dotted lines.

**Figure 7 ijms-21-09756-f007:**
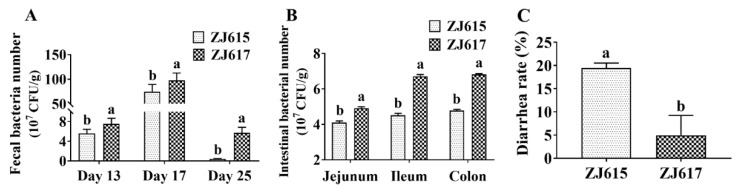
Adhesion of *L. reuteri* ZJ617 and ZJ615 in the porcine intestine. The number of lactobacilli in discharged feces and homogenized intestinal segments was determined by serial dilution on agar plates in the presence of streptomycin. (**A**) Number of bacteria in fecal samples. Feces were collected on days 13, 17, and 25. (**B**) Number of bacteria in the intestine after piglets were sacrificed at 25 days. (**C**) Diarrheal rate, calculated as the product of the number and times of diarrhea piglets divided by the product of the number of piglets raised and days in the experimental period. A one-way ANOVA with Duncan’s multiple range test was used (*n* = 6). Different letters indicate significant differences between the two strains (*p* < 0.05).

**Table 1 ijms-21-09756-t001:** Binding energy in the interaction of *L. reuteri* GAPDH with different mucin glycans and key amino acid residues.

Glycan	Interacting Amino Acids	H-bond Distance (Å)	Binding Energy (BE, kcal/mol)
GalNAcα1-O-Ser	MET1	1.9, 2.1	−3.43
	SER2	/	
	VAL3	3.2	
	SER29	3.1, 2.8, 1.9	
	ASP30	2.9	
	ILE31	/	
	LEU335	/	
	ALA338	/	
	THR339	/	
GalNAcαSer	MET1	2.0, 2.2	−3.24
	SER2	/	
	VAL3	/	
	LYS27	2.8	
	SER29	/	
	ASP30	2.1, 1.8	
	ILE31	/	
Galβ3GalNAc	MET1	2.1	−3.33
	SER2	/	
	VAL3	/	
	SER29	1.9, 2.0	
	ASP30	2.0, 2.2, 2.4, 4.0	
	ILE31	/	
	LEU334	/	
	LEU335	/	
	LYS336	/	
	ALA338	/	
	THR339	/	

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
