# Peer review of "Glyceraldehyde-3-Phosphate Dehydrogenase Increases the Adhesion of *Lactobacillus reuteri* to Host Mucin to Enhance Probiotic Effects"

_ijms, 2020, doi:10.3390/ijms21249756_

Round 1

Reviewer 1 Report

The manuscript shares interesting facts. With the increase in the use of antibiotics the use of probiotics which maintains the balance of gut microbiota is also very important . Many organisms are gaining importance as probiotic , the one with better adherence property wins the race. Here the author have shown a well defined relation between the high level of GAPDH leading to adhesion and cell permeability , two very important factors in probiotic selection.

Author Response

Thank you for your comments.

Reviewer 2 Report

The authors clearly introduce the topic of the importance of adhesion in probiotic bacteria and the role of adhesions in this process along with the key protein GAPDH for adhesion of Lactobacillus strains to mucins in the gut. The objectives of the study to investigate the secretion and distribution of GAPDH on the cell surface and determine tole in adhesion to mucin were supported by the experimental approaches using flow cytometry for permeability analysis and protein modeling using (SWISS-MODEL).

Line 44 – cell implied – or alternatively use strain ATCC 53608, punctuation is needed in this run-on sentence to clarify which adhesion proteins are found associated with each strain mentioned in the text.

Line 62 – reference for sugar-binding protein interactions between gut bacteria and mucins needed here

Lines 92 - 93 – when strains were adjusted to the same concentration (cells/ml??) need clarification/definition by authors – unclear from M7M section how this adjustment was made. More details on how the cultures were adjusted based on growth curves need to be included.

Figure 1B – while it is clear that the low adhesive strain has less GAPDH on the surface compared to high adhesive strain - the letters representing significant differences is not defined – these numbers need to be included or defined by authors - grayscale value need units or relative numbers with the definition

Line 124 – second (A) should be (B) – flow cytometry for ZJ6117

Units for 2 (D-F) – if arbitrary – indicate here

Line 131 – sentence structure – awkward - rephrase

Figure 3 – arrows pointing readers to where authors are making the claim that GAPDH is re-localizing with time is needed. Images appear slightly out of focus making it difficult to agree with the author's interpretation of the protein localization.

Figure 4 – (B) letters indicating significance need the definition/define adhesion number (per 100 cells per cm2?)

Figure 5 – units for y-axis - - a key embedded in the figure would be easier for the reader to follow experimental variables rather than in legend with obscure wording then the reader has to go back to figure to ascertain results,

Section 2.6 Molecular Interaction between….line 183 – if this conclusion is based on the author's modeling move the conclusion after the presentation of results. If this is previous knowledge from other interactions between GAPDH and mucin glucans then a reference is needed.

Figure 7 – define letters – why would higher levels of strain ZJ617 in feces indicate higher adhesion relative to 15 – significant excretion would seem to mean less adhesion, no? Clarification of experiment and adhesion high levels needed.

Discussion –

Paragraph beginning line 225 – again explain the significance of excreted high levels of strain 617 and the relationship to adhesion. Unclear and not explained.

Line 249 – authors finally explain the adjustment of cells with units – this needs to be moved to the results section for clarification.

It would be a cleaner discussion if the authors indicated the figures they are discussing in the text. It makes for easier reading if the reader wants to compare results with discussion points.  For example, Line 268 We used the Octet system (Figure 5)  to determine in vitro time-resolved interaction dynamics between lactobacilli and mucin, revealing a high affinity between the two – OR -  (Figure 5).
